# Chloride Diffusion by Build Orientation of Cementitious Material-Based Binder Jetting 3D Printing Mortar

**DOI:** 10.3390/ma14237452

**Published:** 2021-12-04

**Authors:** Kyung-Sung Min, Kwang-Min Park, Bong-Chun Lee, Young-Sook Roh

**Affiliations:** 1Construction Technology Research Center, Korea Conformity Laboratories, Construction Division, Seoul 08503, Korea; mks2523@kcl.re.kr; 2Metal Machinery Center, Korea Conformity Laboratories, Components & Materials Division, Incheon 21591, Korea; leebc@kcl.re.kr; 3Department of Architectural Engineering, Seoul National University of Science and Technology, Seoul 01811, Korea; rohys@seoultech.ac.kr

**Keywords:** additive manufacturing, binder jetting 3D printing, additive direction, durability, compressive strength, chlorine diffusion

## Abstract

Binder jetting 3D printing (BJ3DP) is used to create geometrical and topology-optimized building structures via architectural geometric design owing to its high degree of freedom in geometry implementation. However, building structures require high mechanical and durability performance. Because of the recent trend of using 3D printing concrete as a structural component in reinforcing bars, its durability with respect to chloride penetration needs to be reviewed. Therefore, in this study, the compressive strength and durability of the chloride diffusion of cement-based 3D-printed output were evaluated. In addition, to confirm the performance difference based on the build orientation, the compressive strength and chloride diffusion were evaluated with respect to the build direction and transverse direction. The experimental results show that the compressive strength was approximately 22.1–26.5% lower in the transverse direction than in the build direction and that the chloride diffusion coefficient was approximately 186.1–407.1% higher in the transverse direction. Consequently, when a structure that requires long-term durability is produced using BJ3DP, it is necessary to examine the design and manufacturing methods in relation to the build orientation in advance.

## 1. Introduction

### 1.1. 3D Printing Technology for Construction

Additive manufacturing (AM), which is also referred to as three-dimensional (3D) printing, is a concept that contrasts with subtractive manufacturing (SM), where material production is realized by cutting or trimming [1]. In ASTM F 2792-12 (2015), AM is defined as “a method of stacking continuous materials layer-by-layer to create geometry from 3D model data, which is a technology that contrasts SM [2,3,4]”.

The application range of 3D printing technology and materials has been extended to daily necessities, machinery, electronics, and medicine. Experts in these fields have collaborated closely with construction technology engineers to create high-value- construction applications using 3D printing for geometries that are difficult to implement with existing methods, such as atypical members and geometry optimization [5,6].

Representative 3D printing technologies used in the construction field include material extruded 3D printing (ME3DP) and binder jetting 3D printing (BJ3DP). ME3DP is a method of extruding construction materials, such as mortar and concrete, from a nozzle using pressure. It is mainly used on-site for the construction of large members such as columns and walls owing to its advantages of easy on-site 3D printing and enlargement [7]. BJ3DP is a method of forming 3D structures through bonding between powder particles by discharging a liquid adhesive on a powder-type material. Compared with ME3DP, it has the advantage of implementing geometrical and topology-optimized structures because of its high degree of freedom in the geometry of architectural construction [8,9,10].

The structures printed using a 3D printer must meet specific mechanical performance requirements. It has been reported that the materials used and the build orientation affect the mechanical performance of the 3D output [11,12,13,14]. Several studies have been conducted examining the mechanical performance of the output and its relation to the build orientation, but they are limited to the evaluation of short-term performance, such as compressive strength [15,16], tensile strength [17,18], flexural strength [19], and shear strength [20].

The printability and mechanical properties of 3D printing concrete have been widely investigated in laboratories. Furthermore, optimizations are being performed to improve the mechanical properties of 3D printed concrete, including the interlayer strength. For this purpose, mesh reinforcing [21] and 3D concrete printing with a reinforcing bar [22], fibers [23], or admixtures [24] have been employed.

### 1.2. Binder Jetting 3D Printing

A number of studies examined physical performance based on the material. For example, Gibbons et al. used rapid hardening cement (RHC) [25], Maier et al. used calcium aluminate cement [26], and Cesaretti et al. used magnesium oxide cement [27] for BJ3DP. The list of materials that can be used for BJ3DP has been continuously expanding.

In this study, ProJet CJP 360 from 3D Systems was used. The build volume of the device is 203 mm × 254 mm × 203 mm, and two to four layers can be deposited per minute with a thickness of 0.089–0.102 mm for each layer. BJ3DP deposits powder by injecting a liquid binder into a powder bed [28,29] and draws a 2D pattern by applying a layer of powder to a build plate and injecting the liquid adhesive binder into specific parts [30,31].

A schematic of the BJ3DP process is illustrated in Figure 1a [32]. The setting of the 3D printer is spread over the bed surface of the powder by the leveling roller corresponding to the thickness of the layer (approximately 0.1 mm). Subsequently, a print head jets the liquid adhesive binder to the powder bed to create a 2D pattern on the layer. The binder droplets thus formed are selectively applied to the powder layer, thus binding the powder particles with each other (Figure 1b). After each layer is spread, the build piston is lowered to accommodate the next layer, and the process is repeated.

After the 3D output is obtained by repeating this process, the unbound powder is removed. When cementitious materials are used in BJ3DP, performance improvement after processing is essential because the mechanical performance immediately after printing is low [33,34,35]. Figure 2 shows the binder jetting 3D printing process and postprocessing for strength improvement. Figure 2a–c shows the binder jetting 3D printing process; Figure 2a shows the selectively applied binder to the powder surface in a specific layer, whereas Figure 2b shows the completed 3D printing output until the final layer. Furthermore, Figure 2c shows the output after depowdering. Figure 2d–g shows the postprocessing for strength improvement. The test specimens were manufactured through vacuum impregnation, temperature curing at 70 °C, and water curing.

### 1.3. Introduction of Chloride Diffusion for Durability Evaluation

Steel-reinforcing-bar corrosion caused by chloride penetrating concrete is recognized as the major factor that causes the deterioration of concrete structures [36]. Reinforcing bar corrosion causes concrete cracks and cladding, resulting in a significant reduction in the strength of the structure [37,38].

A technology for inserting reinforcing bars into 3D printed concrete has recently been developed [39,40]. An experimental study is therefore needed to confirm the corrosion of the reinforcing bar caused by chloride. In particular, it is reported that 3D printed concrete is mainly implicated in the differential degradation of durability depending on the build orientation [41]. Chloride penetration in MD3DP concrete has been investigated in previous studies [42,43]. However, BJ3DP concrete has not been studied thus far. Therefore, this study aimed to evaluate the chloride penetration in BJ3DP according to the build orientation.

### 1.4. Research Objectives

Recently, 3D printer output has been used as a building member. However, methods and research results for evaluating the durability of printouts have not been reported. To utilize the cementitious material-based 3D printing output as a building member for long-term use rather than a prototype for temporary use, its long-term durability performance should be examined in addition to its short-term mechanical performance. As a step in this direction, the chloride diffusion coefficient was evaluated in this study using NT build 492 [44], a representative method for evaluating the durability of cementitious materials. Additionally, the influence of the build orientation (build and transverse directions) on chloride diffusion was evaluated. Figure 3 shows a schematic of the research plan and the purpose of this study.

## 2. Materials and Methods

### 2.1. Materials Used

#### 2.1.1. Powder and Binder

For cementitious materials used for binder jetting, rapid reactions and hardening are essential when an adhesive is injected [45,46,47]. Therefore, in this study, alkali-activated materials (AAMs) and RHC were used to rapidly harden through reactions with water and develop the required strength after printing. Table 1 lists the physical properties and chemical compositions of the materials used. According to the literature, the injection of VisiJet^®^ PXL, an adhesive for ProJet CJP 360 [48], onto cementitious materials further decreases the compressive strength, compared with the injection of ordinary distilled water as an adhesive [32,49]. Thus, ordinary distilled water was injected as an adhesive in this study.

#### 2.1.2. Mixture Design

In a previous study [32], a basic methodology was developed for alkali-activated materials (AAM)-based BJ3DP. We used AAM comprising ground granulated blast-furnace slag (GGBFS) and fly ash (FA) as the major components of the BJ3DP powder. The AAM powder was synthesized using an optimal mixture ratio described in a previous report [32]. Silica sand with a size of 0.1–0.17 mm was mixed with AAMs and RHC. BJ3DP mortar specimens were printed using a binder (powder) with a silica sand ratio of 0.75:0.25. Table 2 lists the binder compositions.

#### 2.1.3. Postprocessing

Compressive strength specimens (20 mm × 20 mm × 20 mm) and durability (ϕ100 mm × 50 mm) specimens were produced using BJ3DP, and postprocessing was performed for strength improvement. A post-storage solution was prepared by mixing liquid sodium silicate (SiO_2_ 28.2%, Na_2_O 9.3%, and H_2_O 65.5%) and pure 98% sodium hydroxide (NaOH) (Na_2_SiO_3_/NaOH ratio of 4 and 3 mol of NaOH). Table 3 shows the composition of the mixed poststorage solution. The poststorage solution was prepared using the optimal mixture ratio reported in the previous study [32].

Postprocessing was performed using the following method [37,38]. The outputs were printed and dried for 2 h in a powder bed. After 24 h, depowdering was carried out using a compressed air gun to remove any unbound powder. Subsequently, a basic postprocessing procedure was carried out as follows: (1) The BJ3DP output was immersed in the postprocessing storage solution in a vacuum impregnator. (2) A maximum pressure of 0.10 MPa was maintained, and the postprocessing solution permeated the voids inside the printed output. (3) Vacuum impregnation was continued until no air bubbles were produced in the printed output, which in this case required ~10 min. (4) The output was immersed in the postprocessing storage solution in a temperature chamber at 70 °C for 7 days. (5) The output was removed from the postprocessing solution and wiped off with distilled water to remove any remaining solution from its surface. (6) After curing in a 70 °C temperature chamber for 7 days, water curing (20 ± 2 °C) was carried out for 28 days [32].

### 2.2. Experimental Method

#### 2.2.1. Compressive Strength

To examine the influence of the build orientation on the mechanical performance of the 3D printer outputs, loads were applied in the build and transverse directions, as shown in Figure 4. Compressive strength specimens were fabricated through the postprocessing procedure described in Section 2.1.3 after printing cubic specimens with a size of 20 mm × 20 mm × 20 mm. Compressive strength was tested according to the Korean Industrial Standards of KS L ISO 679 [50] using a universal testing machine (UTM, Instron Universal Testing Machine, MA, USA; maximum load of 50 kN) at 28 days of age. Compressive strength was measured for five specimens of each material in both directions.

#### 2.2.2. Durability (Chloride Diffusion)

To examine the influence of the build orientation on the durability performance of the 3D printer outputs, a chloride diffusion test was conducted in the build and transverse directions, as shown in Figure 5. In this study, the chloride diffusion coefficient was evaluated by NT build 492. This coefficient is the chloride migration coefficient from nonsteady state migration experiments [44], in which an electrically accelerated test method is applied to the DuraCrete model, a representative method to examine the durability of cementitious materials. For the durability test, specimens with a size of ϕ100 mm × 50 mm were fabricated through the postprocessing procedure described in Section 2.1.3 after printing. The durability was measured using three specimens for each material and direction.

As pretreatment, the specimens were subjected to vacuum saturation (0.10 MPa) for 3 h and then immersed in a saturated solution of calcium hydroxide (Ca(OH)_2_) for 18 ± 2 h. In the electrically accelerated test, an electrical potential difference was applied by filling the anode (+) with a 0.3 M NaOH aqueous solution and the cathode (−) with a 10% NaCl aqueous solution. The secondary voltage was calculated by applying an initial voltage of 30 V and measuring the current value for the initial voltage. Consequently, a test time of 6 h was determined based on the current value. Upon completion of the test, each specimen was split, and a 0.1N silver nitrate (AgNO_3_) aqueous solution was applied. The silver chloride line was then measured using a digital Vernier caliper, and the average value was calculated. The chloride diffusion coefficient was calculated using Equation (1):(1)$$D_{nssm}=\frac{0.0239\left(273+T\right)L}{\left(U-2\right)t}\left(x_{d}-0.0238\sqrt{\frac{\left(273+T\right)Lx_{d}}{U-2}}\right)$$
where $D_{nssm}$ is non-steady-state migration coefficient, ($\times 10^{-12}\;m^{2}/s$), U is the absolute value of the applied voltage (V), T is the average value of the initial and final temperatures in the anolyte solution (°C), L is the thickness of the specimen (mm), $x_{d}$ is the average value of the penetration depths (mm), and t is test duration (hour).

## 3. Results

### 3.1. Compressive Strength Test Results

Figure 6 shows the compressive strengths of the BJ3DP specimens with respect to the build orientation at 28 days of age. Figure 6a shows the compressive strength of the AAM. The measurement results show that the compressive strength was approximately 22.1% lower in the transverse direction, as 25.8 MPa was observed in the build direction and 20.1 MPa in the transverse direction. Figure 6b shows the compressive strength of the rapid hardening mortar (RHM). The compressive strength was approximately 26.5% lower in the transverse direction than in the build direction, as 18.9 MPa was observed in the build direction and 13.9 MPa in the transverse direction.

It appears that the strength was reduced in the transverse direction because gaps between interlayers were generated in the BJ3DP deposition process. As a result, the mechanical properties of BJ3DP are influenced strongly by the printing direction. Compared with conventional concrete or mortar, the critical problems of the mechanical properties of 3D printed concrete or mortar are the interlayer bond strength and anisotropy. The weak interface bond leads to the reduction of the mechanical properties and durability of 3D printed concrete or mortar [51]. Therefore, further research should be conducted in the future to improve the interlayer bond strength of BJ3DP.

### 3.2. Chloride Diffusion Test

Figure 7 and Figure 8 and Table 4 show the chloride diffusion coefficient at 28 days of age after the mortar specimens (∅100×50 mm) printed using BJ3DP were subjected to postprocessing and curing. In general, the penetration resistance increased as the chloride diffusion coefficient decreased.

After splitting the specimens tested for 6 h under an applied voltage of 10 V, 0.1 N silver nitrate (AgNO_3_) was applied to the specimens. The results are shown in Figure 7. Chloride ions penetrated and reacted with silver nitrate at depths of 2.58–3.30 mm (Figure 7a), 6.64–8.13 mm (Figure 7b), 4.88–5.54 mm (Figure 7c), and 7.99–8.63 mm (Figure 7d). The measured chloride penetration depths were substituted into Equation (1). The obtained results are shown in Table 3.

In the case of the AAM shown in Figure 8a, chloride diffusion coefficients of 8.0–10.6 × 10^−12^ m^2^/s and 29.1–37.9 × 10^−12^ m^2^/s were measured in the build, and transverse directions, respectively, and the average diffusion coefficient in the transverse direction was 407.1% higher than that in the build direction. In the case of the RHM shown in Figure 8b, chloride diffusion coefficients of 19.3–22.7 × 10^−12^ m^2^/s and 37.3–40.9 × 10^−12^ m^2^/s were measured in the build and transverse directions, respectively. The average diffusion coefficient in the transverse direction was 186.1% higher than that in the build direction.

It appears that the diffusion coefficient in the transverse direction was higher because gaps between layers were generated in the BJ3DP deposition process, and the gaps facilitated the chloride penetration. When printing was performed in the build direction, chloride diffusion decreased because the output direction was perpendicular to the chloride diffusion direction. However, when printing was performed in the transverse direction, chloride diffusion increased compared with that in the build direction because the output direction was parallel to the chloride diffusion direction. Nerella et al. [51] also observed that the interface exhibits long and wide separations between the two neighboring layers.

## 4. Results and Discussion

In this study, the performance of the specimens produced using the binder jetting 3D printing (BJ3DP) method was evaluated with respect to the build orientation. The difference in performance depending on the build orientation was examined through the compressive strength and durability (chloride diffusion) tests. The conclusions drawn from this study are summarized as follows:The compressive strength of the BJ3DP outputs was 22.1% to 26.5% lower in the transverse direction than in the build direction.The chloride diffusion coefficient in the transverse direction was 186.1% to 407.1% higher than that in the build direction. Chloride diffusion appeared to decrease when printing was performed in the build direction because the output direction was perpendicular to the chloride diffusion direction. In contrast, it appeared to increase when printing was performed in the transverse direction because the output direction was parallel to the chloride diffusion direction.

In this study, the build orientation was limited to two types: build and transverse directions. The specimen curing age was also limited to 28 days. In the future, we will investigate the diversification of the build orientation (e.g., 15, 30, 45, 60, and 75°) and the long-term curing age after 28 days.

## 5. Conclusions

The influence of the output direction on the compressive strength ranges from 22.1% to 26.5%, whereas its influence on the durability performance ranges from 186.1% to 407.1%. This shows that the influence of the output direction on the long-term durability performance is greater than that on the short-term mechanical performance. Therefore, it is necessary to examine the output direction that can secure durability performance in advance for the long-term use (considering maintenance) of cementitious material-based 3D printing outputs. Therefore, when the BJ3DP outputs are produced based on cementitious materials, the initial design and manufacturing methods should be determined by examining the self-weight and external force directions in advance.

In this study, cementitious material-based outputs were additively manufactured using BJ3DP. Because AM exhibits different mechanical properties depending on the build orientation, the build orientation needs to be considered during design. It is also necessary to examine the build orientation in advance to satisfy the performance required for the use of cementitious material-based outputs as building members in the future. In further research on the design and manufacturing of building members using BJ3DP, we intend to set the output size, geometry, and material used as the research variables.

The separation of layer interface of BJ3DP caused by the loose microstructure with more and larger pores leads to the degradation of durability, and it will be considered in future work. Considering the importance of durability to the service life of concrete structures, research on the durability of BJ3DP needs to be emphasized.

## Figures and Tables

**Figure 1 materials-14-07452-f001:**
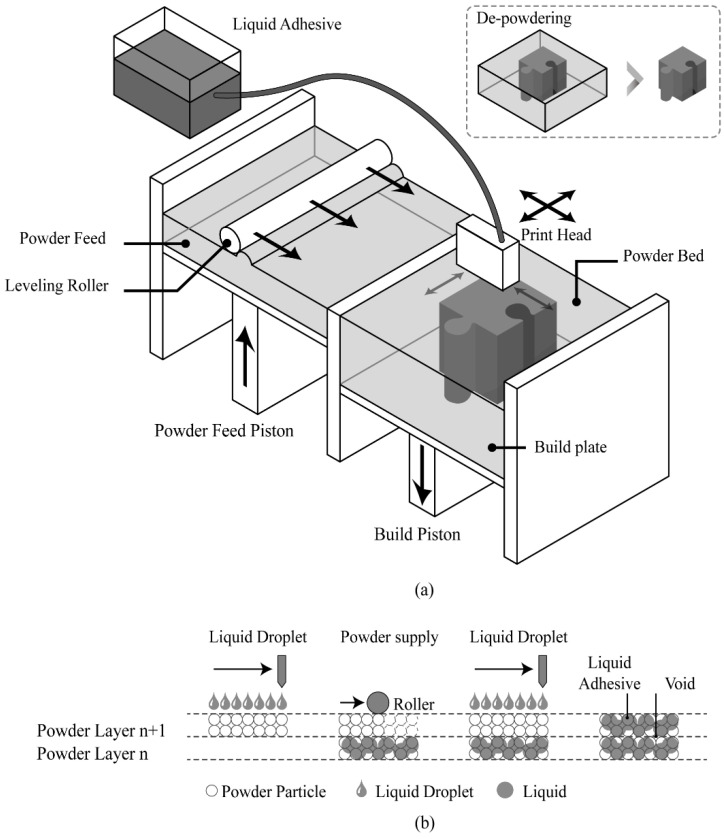
Schematic illustrations of binder jetting 3D printing (BJ3DP): (**a**) BJ3DP system and (**b**) powder/binder interaction between adjacent layers. Adapted with permission from ref. [32]. Copyright 2021 Kwang-min Park.

**Figure 2 materials-14-07452-f002:**
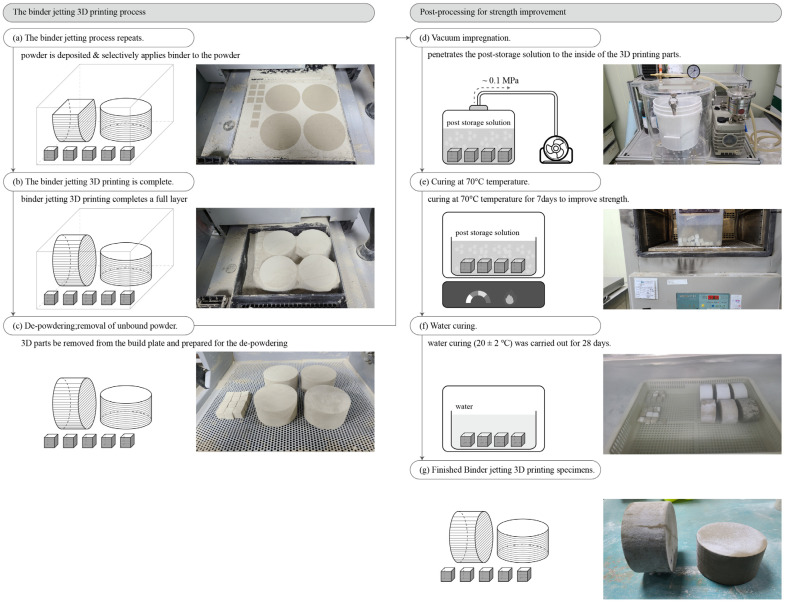
Binder jetting 3D printing process and postprocessing for strength improvement: (**a**) The binder jetting process repeats, (**b**) The binder jetting 3D printing is complete, (**c**) De-powdering removal of unbound powder, (**d**) Vacuum impregnation, (**e**) Curing at 70 °C temperature, (**f**) Water curing, and (**g**) Finished Binder jetting 3D printing specimen.

**Figure 3 materials-14-07452-f003:**
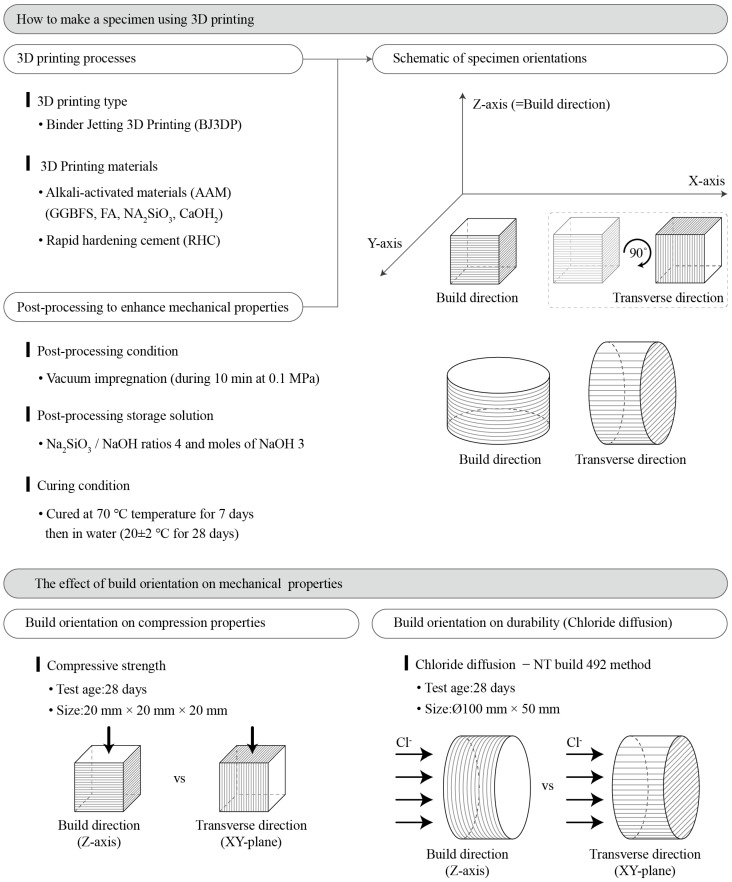
Research plan and purpose.

**Figure 4 materials-14-07452-f004:**
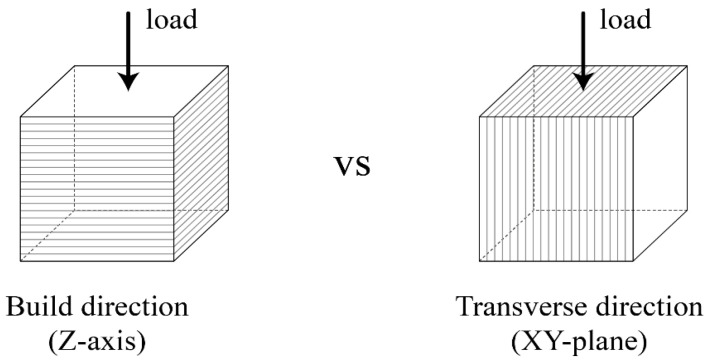
Loading for compressive strength considering build orientation.

**Figure 5 materials-14-07452-f005:**
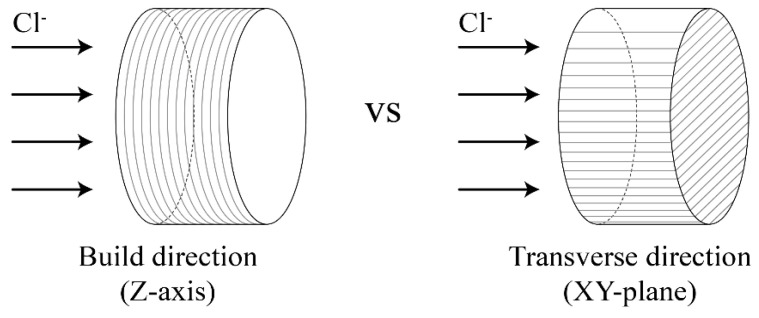
Chloride penetration considering the build orientation.

**Figure 6 materials-14-07452-f006:**
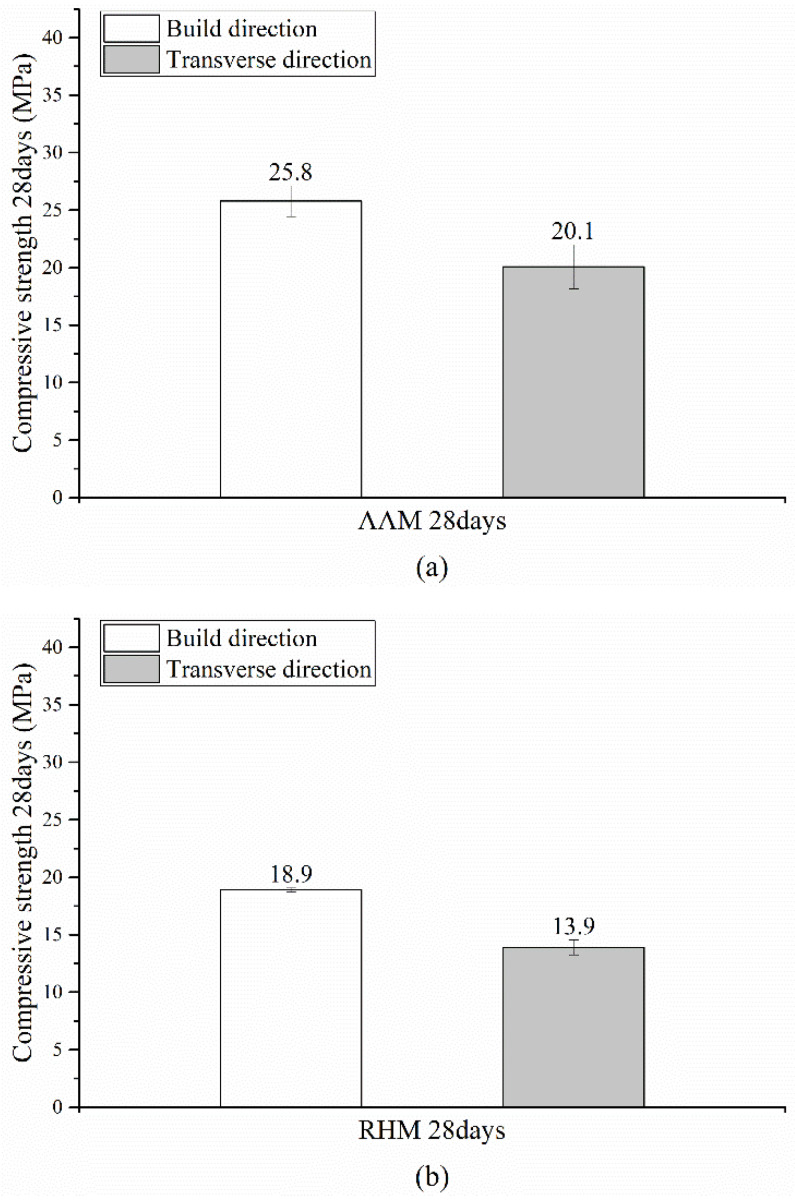
Compressive strength test results with respect to the build orientation: (**a**) alkali-activated mortar (AAM) and (**b**) rapid hardening mortar (RHM).

**Figure 7 materials-14-07452-f007:**
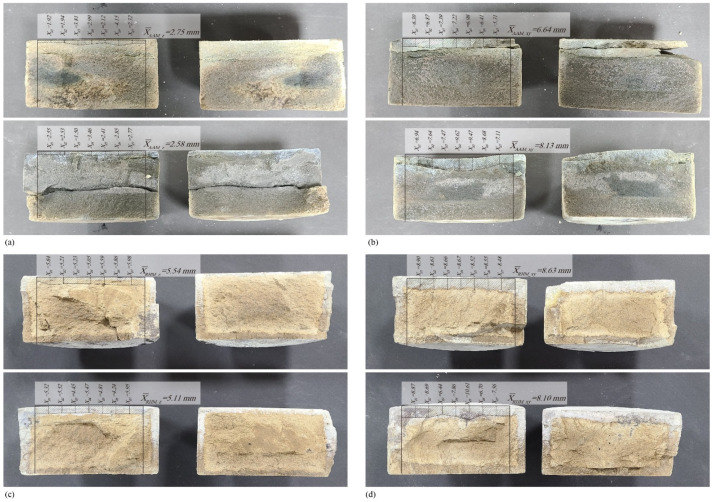
Chloride diffusion test results: (**a**) alkali-activated material (AAM) build direction, (**b**) AAM transverse direction, (**c**) rapid hardening mortar (RHM) build direction, and (**d**) RHM transverse direction.

**Figure 8 materials-14-07452-f008:**
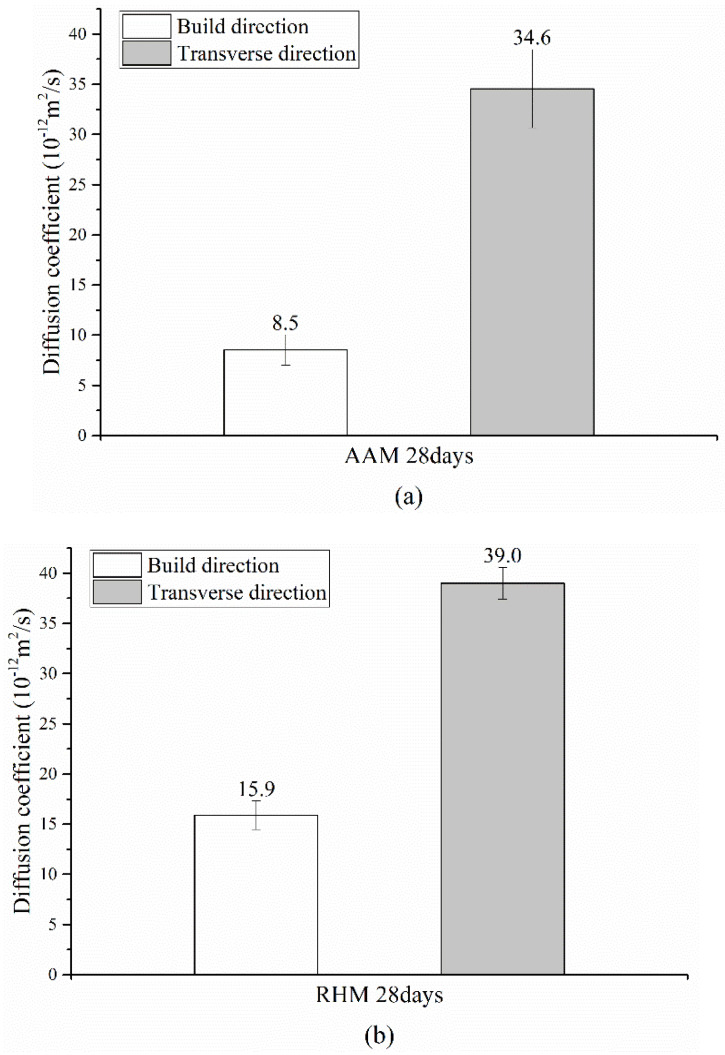
Chloride penetration resistance test results with respect to the build orientation: (**a**) alkali-activated material (AAM) and (**b**) rapid hardening mortar (RHM).

**Table 1 materials-14-07452-t001:** Physical properties and chemical compositions of the powders used in binder jetting 3D printing (alkali-activated material (AAMs); rapid hardening cement (RHC)).

Type	Chemical Compositions (wt%)										Physical Properties	
	CaO	Al_2_O_3_	SiO_2_	Fe_x_O_y_	MgO	TiO_2_	Na_2_O	K_2_O	SO_3_	Lg. Loss	Density (g/cm^3^)	Surface Area (cm^2^/g)
AAMs	31.50	11.70	38.40	1.30	1.77	0.46	4.54	0.67	2.04	7.62	2.25	5460
RHC	45.14	22.02	10.90	3.88	1.08	1.01	0.29	0.59	14.89	0.20	2.89	5700

**Table 2 materials-14-07452-t002:** Mixture design for binder jetting 3d printing (alkali-activated mortar (AAM), rapid hardening mortar (RHM)).

Type	Unit Weight (g)					
	AAMs ^1^				RHC ^2^	Silica Sand
	GGBFS	FA	Na_2_SiO_3_	Ca(OH)_2_		
AAM	4642	1161	1044	653	-	2500
RHM	-	-	-	-	7500	

^1^ AAMs: alkali-activated materials consisting of ground granulated blast furnace slag (GGBFS), fly ash (FA), Na_2_SiO_3_, and Ca(OH)_2_. ^2^ RHC: rapid hardening cement.

**Table 3 materials-14-07452-t003:** Composition of postprocessing materials.

Postprocessing Materials	Unit Weight (g)		
	Na_2_SiO_3_	NaOH	Water
Na_2_SiO_3_/NaOH ratio of 4 and 3 mol of NaOH	1280	120	200

**Table 4 materials-14-07452-t004:** Chloride diffusion coefficient depending on the BJ3DP material and the build orientation (alkali-activated mortar (AAM), rapid hardening mortar (RHM)).

Type	Direction	Average Chloride Penetration Depth (mm)	Diffusion Coefficient (10^−12^ m^2^/s)	Standard Deviation
AAM	Build direction (Z-axis)	2.75	8.0	1.51
		2.58	7.0	
		3.30	10.6	
	Transverse direction (X-Y Plane)	6.64	29.1	3.89
		8.13	37.9	
		7.86	36.7	
RHM	Build direction (Z-axis)	5.54	22.7	5.60
		5.11	20.4	
		4.88	19.3	
	Transverse direction (X-Y Plane)	8.63	40.9	4.71
		8.10	37.9	
		7.99	37.3	

## Data Availability

The data presented in this study are available from the corresponding author.

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
