# Peer review of "Chloride Diffusion by Build Orientation of Cementitious Material-Based Binder Jetting 3D Printing Mortar"

_materials, 2021, doi:10.3390/ma14237452_

Round 1

Reviewer 1 Report

The manuscript entitled "Durability Performance due to Additive Direction in Cementitious Materials—Binder Jetting 3D Printing" presents an experimental study conducted on the obtaining of two types of materials (an alkali-activated material and a cement-based material) by 3D printing. Moreover, the study evaluates the compressive strength and the chlorine diffusion for the 56 days aged samples. However, the scientific organization of the paper is questionable as there is no clear purpose of choosing the specific amount of each component from the AAM, the number of tests is very limited, the number of the tested specimen isn’t provided (a deviation bar was presented on the bars from Figure 6, however, the number of tested samples should also be specified) and many other issues must be addressed. The introduction section doesn’t provide a clear overview regarding the obtaining of cementitious materials by 3D and the currently used methods/parameters (state of the art), also other analyses should be performed to correlate the obtained results. There are no discussions (comparative study with existing literature, citations to support the observed behavior, etc.) in the results section.

The title should be changed, to highlight the durability performance a time depending experiment should be conducted. In other words, the same analysis should be realized at different samples ages, starting with 7 days, followed by 14, 56, 90 days, etc. This study only evaluates the compressive strength and Chloride diffusion coefficient of 56 days of aged samples obtained by 3D.

The paper needs major revisions before being processed further.

Abstract

There is no clear purpose of the study in the abstract. Please highlight the novelty of the study.

Introduction section

The introduction section must be significantly improved.

In the introduction section, a comprehensive and exhaustive review of the state of the art in the field of the study must be provided. Please refer to previous works, and highlight the experiments and results published previously. In the current form, the introduction section only provides basic/general information about 3D printing.

Also, the novelty of this study isn’t presented. Please include a sentence, in the last paragraph from the introduction section, to highlight the novelty of this study.

"However, methods and research results for evaluating the durability of printouts were not reported." - please reevaluate the literature. See previous results published in https://doi.org/10.1016/j.istruc.2020.12.061, https://www.jstor.org/stable/j.ctv13xpsvw.41 or https://doi.org/10.1016/j.cemconres.2019.105780

Moreover, The citations have been introduced in bulk form “[1-4], [11-14], [25-30]” and not distributed in the text in accordance with the affirmations that must be supported. Also, to avoid this type of citation, you can cite review studies. Please introduce citation at a specific position to assure a clear correspondence between the affirmations from the introduction section and the previous publication.

Figure 2. The author states that (a) represents 3D printing, however, this is a method, it cannot be represented in a picture/photo. Maybe (a) represents the build plate. Please make corresponding corrections. Also, (b) de-powdering, the photo seems to show the samples before de-powdering, etc.

Moreover, in the text Figure 2 was introduced as: "Figure 2 shows the process of fabricating test specimens using 63 BJ3DP", however, it can be seen that Figure 2 doesn’t show the process, it only shows the evolution of samples after each stage. Please improve the description of Figure 2.

Figure 3. The author states that "Cured at high temperature 70 °C"- this temperature cannot be considered high, please replace the term "high" with a more suitable one, maybe "medium".

Materials and Methods section

Table 1 - - two types of iron oxides have been detected in ash (see DOI: 10.3390/ma13143211), therefore, please replace Fe2O3 with FexOy.

The sum of the components from AAMs and RHC isn’t 100, please check your experimental results.

Table 2 – the sum of the components from AAM isn’t 7500, therefore, the above-mentioned ratio wasn't assured, please check your calculations.

" The output was removed from the post-processing solution and sprayed with distilled water to remove any remaining solution from its surface." How could spraying with distilled water assure the removal of the post-processing solution? Does spraying refer to the washing or is this method also described in a previous publication?

"The compressive strength of the specimens was measured at a rate of 10 mm/min .... at 28 days of age." How was the measuring rate established? Please refer to the standard or previous studies.

Results section

The research presents the experimental results with very limited discussions. The obtained results should be compared with the literature, to highlight the findings from this study.

"exhibited excellent compressive strength" – compared to what? Please cite previous studies or refer to standard concrete.

Figure 7 – please introduce figure labels to improve the scientific value of the images. Please introduce labels with indicating the area of interest for the reader.

Discussion section

Please rename this section as "Conclusion" – in the discussion section a comparison between the obtained results and the results obtained by other authors should be presented

Currently, the discussion section includes the results of the study presented in a long and unclear manner.

This section should be improved.

Author Response

Response has been uploaded

Reviewer 2 Report

The manuscript discusses the utilisation of binder jetting 3D printing and compares the performance of specimens by means of compressive strength and durability in two directions, namely the build and the transverse one. In general, the manuscript lacks a more scientific discussion of the results and could benefit from some interpretation of them. Moreover, terms like ‘we’ in Line 275 are recommended to be avoided to obtain objectivity for the potential reader. The English language level is good.

More specific comments are given below:

The title can be improved as now does not describe in the best way the topic. Moreover, the second part of the title seems not to be smoothly linked with the first part.

Figure 1 & 2: The figures need to be explained in the body of the text for all the different steps and parts included. The authors may need to clarify if Fig 1 is an authors’ creation; otherwise, the source reference should be included in the figure’s label.

Line 92: Please provide references for this method.

Line 139: How the processing solution was ensured and inspected? Please clarify.

Line 153-154: Please include the protocol followed of the standardised test for the calculation of the compressive strength.

Line 179: Is this the scientific way to measure the chloride line? Image detection and digital image processing would assist to increase precision.

Sections 3.1 & 3.1 & conclusions: This is the main concern of the manuscript. The authors need to extensively discuss the reasons for the results besides the reason for the gaps between the layers rather than a numerical presentation.

Table 4: Please provide the coefficient of variation or standard deviation of the replicates.

Author Response

Response has been uploaded

Reviewer 3 Report

Dear authors;
I sincerely congratulate you on the quality of your research.

In this article (materials-1428663), the research on the impact of different 3D printing directions on the durability of the building structure is carried out.

The 3D printing technology was introduced in the abstract and the research object was clearly put forward; the performance of 3D concrete was tested; the experimental results were analyzed, and some meaningful conclusions were drawn; and the follow-up research plan was also given .  

I think this is an interesting research paper. I hope that the author will continue to conduct in-depth research and get more meaningful research conclusions.

Best

Author Response

Thank you for your kind review. This study is an experimental study to confirm the effect of additive direction on the compressive strength and chloride diffusion of cement-based 3D printing output. It was confirmed that the build direction 3D printed output was excellent in terms of compressive strength and chloride diffusion. Based on the results of this study, we will continue to accumulate a database for using 3D printed output in the construction field.

Reviewer 4 Report

1. L159. Please briefly introduce the relationship between durability and chloride diffusion.
2. L187. The characters are mixed and cannot be seen.
3. Please provide the SEM result about the experiment to further illustrate the mechanism like previous research.

Ref.:

Experimental Study on the Influence of Rubber Content on Chloride Salt Corrosion Resistance Performance of Concrete. Materials, 2021, 14(16), 4706.

Effects of cations in sulfate on the thaumasite form of sulfate attack of cementitious materials. CONSTRUCTION AND BUILDING MATERIALS, 2019, 229, 116865. 

4. Figure 7 is fuzzy. Please increase the definition.
5. Please provide the mechanism in Section 4.

Author Response

Response has been uploaded

Reviewer 5 Report

The manuscript "Durability Performance due to Additive Direction in Cementi-tious Materials—Binder Jetting 3D Printing" is quite interesting and presents an innovative and interesting theme to potential readers, but further corrections are needed:

a) The authors already report in the title that the durability test will be carried out, but in the summary which tests are not mentioned? The concept of durability assessment in cementitious materials is broad, I would suggest that the authors clearly restrict (even in the title) which durability assessment will actually be carried out!
b) Fig. 2 can be replaced by an experimental flowchart, showing through arrows the process of making and obtaining parts in 3D;
c) Item 1.3 is confusing, objective in general has no references, as stated, this seems to me more of a justification. The objectives section must be clear, small and objective, this can be subdivided;
d) There is a missing topic on cementitious material durability, evaluated properties and other conditions, remember that the journal is about material properties, not just related to the process itself. I suggest some works, such as: 10.3390/fib8110069 (commented on durability in wetting and drying cycles and analysis parameters in cementitious materials); 10.1016/j.cscm.2021.e00709; 10.1016/j.cscm.2021.e00675.
e) The dosage and mixing process for the specimens must be better explained by the authors;
f) Is the evaluated durability method standardized in any country, or only through studies in the literature?
h) Change the color of the graphics or the error bar, in the black column it is not possible to see!
i) The discussion is not good in light of other works in the literature, why did the authors not perform an SEM of the degraded samples? Or another kind of microstructural analysis??
k) Lack of conclusion?? Authors must enter a conclusion in their work.

Author Response

Response has been uploaded

Round 2

Reviewer 1 Report

The article is suitable for publication.

Reviewer 2 Report

Thank you for addressing all my comments. Good luck with this publication.

Reviewer 5 Report

Manuscript can be accepted